# Contamination and Health Risk Assessment of Polycyclic Aromatic Hydrocarbons in Seasoning Flour Products in Hunan, China

**DOI:** 10.3390/ijerph20020963

**Published:** 2023-01-05

**Authors:** Minghui Cheng, Zhen Tan, Xiwen Zeng, Zhu Liu, Pingyu Liu, Anwar Ali, Huali Qiu, Wenjun Jiang, Hong Qin

**Affiliations:** 1Xiangya School of Public Health, Central South University, Changsha 410078, China; 2Changsha County Comprehensive Testing Center, Changsha 410100, China

**Keywords:** seasoning flour products, PAHs, health risk assessment

## Abstract

Polycyclic aromatic hydrocarbons (PAHs) are persistent organic pollutants with carcinogenic, teratogenic, and mutagenic effects. Dietary intake is one of the significant exposure pathways of PAHs. In this study, gas chromatography-triple quadrupole mass spectrometry (GC-MS/MS) was used to detect 16 priority PAHs listed by the United States Environmental Protection Agency (USEPA) in seasoning flour products distributed in Hunan Province. The consumption of seasoning flour products by the Hunan population was investigated by questionnaire. The results showed that the detection rate of PAHs in seasoning flour products in Hunan Province was 92.41%. Among them, benzo[a]anthracene (BaA), phenanthrene (PHE), fluoranthene (FLA), and chrysene (CHR) were dominant. The total PAHs and benzo[a]pyrene (BaP) contents of soggy seasoning flour product samples were higher than those of crisp samples and chewy samples. The total amount of PAHs in rod-shaped and flaky samples were higher than that in filamentous and granular samples. The margin of exposure (MOE) values of various seasoning flour products and all age groups (children, adolescents, and adults) was much more significant than 10,000. Moreover, the incremental lifetime of cancer risk (ILCR) values of all age groups were below 1 × 10^−5^. The above results indicate that PAHs in seasoning flour products have a relatively low health risk for the Hunan population. Still, it is recommended that susceptible populations (children, adolescents, etc.) should control their intake of flour products.

## 1. Introduction

Polycyclic aromatic hydrocarbons (PAHs) are a group of hydrophobic and lipophilic compounds with two or more fused aromatic rings, mainly derived from incomplete combustion and pyrolysis of organic matter [1]. They have toxic effects on the blood, heart, nerve, and immune systems and are associated with various cancers, including breast cancer, esophageal cancer, lung cancer, and so on, causing severe damage to human health [2,3,4,5]. Due to their teratogenicity, carcinogenicity, mutagenicity, and multiple biotoxicity, 16 PAHs were listed as priority pollutants by the US Environmental Protection Agency (USEPA) [6]. Recent studies have found that dietary exposure accounts for 70–90% of the total exposure to PAHs, and dietary intake has become the major exposure pathway of PAHs in addition to occupational exposure and smoking exposure [7,8].

Seasoning flour product is a kind of instant food with wheat flour as the main raw material, commonly known as spicy strip, produced by high-temperature frying and seasoning. In recent years, seasoning flour products have been widely popular among Chinese consumers for their spicy and delicious taste. They are even increasingly popular in other countries, such as the United States, Japan, and Singapore [9,10]. According to the survey data, the annual sales volume of seasoning flour products has reached tens of billions of packets, and the population between the ages of 10 and 40 are the main consumer group [11]. However, the food safety of seasoning flour products has also attracted widespread attention [9,12]. Due to the processing method of high-temperature frying, seasoning flour products are susceptible to PAHs pollution [13]. There are few studies on the contamination of PAHs in seasoning flour products, and most of the existing studies are limited to BaP or a few other PAHs congeners [14]. However, the European Food Safety Authority (EFSA) indicated that a systematic evaluation of multiple PAHs would be more suitable for assessing PAHs contamination in food [15,16]. It is worth noting that there is no uniform standard for maximum PAHs lever in seasoning flour products. Therefore, this study uses four representative indicators (BaP, PAH2, PAH4, PAH8) to analyze PAHs pollution in seasoning flour products comprehensively, and further evaluate the health risks of PAHs in seasoning flour products under current standards.

Hunan Province is one of the China’s essential producing areas of seasoning flour products [17,18]. In addition, Hunan residents have a habit of eating spicy food, so the consumption level of Hunan residents on seasoning flour products is relatively higher than that in other areas [10]. However, there have been no systematic and in-depth studies on the pollution of PAHs in seasoning flour products and the health risk assessment of residents in Hunan. Therefore, this study intends to conduct a preliminary investigation into this significant but still unclear field.

In this study, the concentration of 16 USEPA priority PAHs in seasoning flour products widely circulated in Hunan Province were detected, and the consumption of seasoning flour products among local residents was investigated. The present study combined the detection results of 16 PAHs and consumption survey results to quantify the daily dietary exposure level of PAHs, and further assess the Margin of exposure (MOE) and the incremental lifetime cancer risk (ILCR) of the Hunan population for long-term consuming seasoning flour products to exposure PAHs, which is novel but of great significance. On the one hand, the cancer risk assessment of the Hunan population is conducive to exploring the potential health risk of PAHs dietary exposure caused by consuming seasoning flour products to the whole population, which contributes to providing a scientific basis for the formulation of maximum allowable daily intake and maximum levels of PAHs in seasoning flour products. On the other hand, quantitative description and difference analysis of PAHs pollution in various seasoning flour products were carried out to provide a reference for optimizing the processing process of food products factories and the purchase choice of consumers.

## 2. Materials and Methods

### 2.1. Sampling Collection and Preparation

In May 2021, 79 seasoning flour products were sampled from food factories, supermarkets, campuses, and stalls in Hunan Province, China, which included all kinds of seasoning flour products in local circulation. The seasoning flour products were classified according to the taste types (soggy: 51 samples, crisp: 12 samples, chewy: 16 samples) and shape types (filamentous: 11 samples, rod-shaped: 25 samples, flaky: 25 samples, granular: 8 samples). Before analysis, all samples were well packed and stored away from light at 4 °C.

### 2.2. Chemical Analysis of the PAHs

The seasoning flour product samples were crushed and mixed with grinder. The crushed samples were weighed to about 5.0 g, then added with 10 mL acetonitrile and 5.0 g QuEChERS salting-out agent (containing 1.0 g sodium chloride (NaCl) and 4.0 g anhydrous magnesium sulfate). After vortex shaking for 2 min, the samples were extracted by ultrasound for 10 min at 500 W (KQ-500B ultrasonic instrument, Kunshan Ultrasonic Instrument Co., LTD., Jiangsu, China). Then, after freezing for 30 min at −20 °C, and centrifuging at 4000 r/min for 3 min at 4 °C (Eppendorf 5804R refrigerated centrifuge, Ebender AG, Co., Ltd., Hamburg, Germany), the supernatant was transferred to a 50 mL centrifuge tube. Repeated extraction was performed following the above process, and the supernatant was combined. 1.2 g QuEChERS dispersive solid-phase extraction reagent I (containing 100 mg N-propyl ethylenediamine, 100 mg C_18_ and 1000 mg neutral alumina) was added to the extraction solution, followed by vortex oscillation for 2 min and low-temperature centrifugation at 4000 r/min for 3 min. Four-milliliters of the supernatant was accurately removed into another test tube, dried with nitrogen at 40 °C, redissolved with 1 mL n-hexane until fully dissolved, and filtered with 0.22 μm organic filter membrane.

The 16 PAHs were measured by gas chromatography-triple quadrupole mass spectrometry (GC-MS/MS) (Agilent 7890B-7000C, Agilent Technologies, CA, USA). Chromatographic separation of PAHs was performed by DB-EUPAH column (30 m × 0.25 mm × 0.25 μm, Agilent Technologies, CA, USA). The 16 PAHs were tested by GC-MS/MS with multiple reaction monitoring modes (MRM), as shown in the study of Li et al. [19]. Each target compound was quantified with a standard external method considering matrix effects. The limit of detections (LODs), the limit of quantifications (LOQs), and recoveries for each target compound in various samples were in the range of 0.04~0.55 μg/kg, 0.12~0.18 μg/kg, and 71.54%~103.7%, respectively.

### 2.3. Analysis of PAHs Pollution

The concentration and detection rates of total PAHs and the 16 PAHs congeners listed by US EPA [20,21] (naphthalene (NAP), acenaphthylene (ACY), acenaphthene (ACE), fluorene (FLU), phenanthrene (PHE), anthracene (ANT), fluoranthene (FLA), pyrene (PYR), benzo[a]anthracene (BaA), chrysene (CHR), benzo[b]fluoranthene (BbF), benzo [k]fluoranthene (BkF), benzo[a]pyrene (BaP), indene [1,2,3-c,d]pyrene (IcdP), dibenz [a,h]anthracene (DahA), benzo[g,h,i]perylene (BghiP) in various seasoning flour products were analyzed. According to the recommendations of the EFSA, BaP, PAH2 (BaP + CHR), PAH4 (BaP + CHR + BaA + BbF), and PAH8 (BaP + CHR + BaA + BbF + BkF + DahA + BghiP + IcdP) were selected as alternative indicators for detection of PAHs pollution [15,22,23]. Based on the detection results of PAHs, a database of PAHs contamination in seasoning flour products in Hunan Province was established. The concentrations and detection rates of total PAHs, individual PAHs homologues, and alternative indicators of PAHs in various samples were calculated. The difference of PAHs concentration in various types of seasoning flour products was detected by one-way analysis of variance or Kruskal-Wallis (K-W) test (*p* < 0.05, the difference was statistically significant).

### 2.4. Seasoning Flour Products Intake Investigation

Self-made questionnaires on the intake of seasoning flour products were distributed to residents in Hunan, China (the questionnaire was presented in the Appendix A). A total of 819 valid questionnaires were collected, including 123 from children (6–9 years), 436 from adolescents (10–19 years), and 260 from adults (20–26 years). The basic personal information, dietary habits, preference, purchase frequency, and single intake of seasoning flour products were collected in the questionnaire. Based on the above information, the database of the consumption of seasoning flour products in Hunan Province was established.

### 2.5. Health Risk Assessment of PAHs

#### 2.5.1. Margin of Exposure (MOE)

Based on the survey results of the consumption of seasoning flour products in Hunan residents, combined with the concentrations of BaP, PAH2, PAH4, and PAH8 in various seasoning flour products, the estimated daily intake (EDI) values of BaP, PAH2, PAH4, and PAH8 of residents in Hunan province were calculated by the following Equation (1):(1)$$\mathrm{EDI}=\frac{C_{i}\times {\mathrm{IR}}_{}\;}{\mathrm{BW}}$$
where C_i_ is the concentration of PAHs indicator i in seasoning flour product (μg/kg); IR is the daily ingestion of seasoning flour product (g/day); BW is the body weight (kg).

The EFSA suggests that risk assessment of genotoxic and carcinogenic contaminants can be performed using the MOE [24,25]. The dietary exposure risks of seasoning flour product were evaluated by calculating the MOE values of four kinds of PAHs pollution alternative indicators (BaP, PAH2, PAH4, and PAH8) using the following formula (2):(2)$$\mathrm{MOE}=\;\frac{{\mathrm{BMDL}}_{10}}{{\mathrm{EDI}}_{i}}\;$$
where BMDL_10_ is lower bound of 95% confidence intervals for 10% additional risk of tumorigenicity (the reference BMDL_10_ values of BaP, PAH2, PAH4 and PAH8 were 0.07, 0.17, 0.34 and 0.49 mg/(kg·BW), respectively) [15]; EDI_i_ is the estimated daily intake of PAHs pollution indicator i (ng/(kg·BW)/d).

#### 2.5.2. Incremental Lifetime Cancer Risk (ILCR)

Taking BaP, a recognized PAHs carcinogen, as the benchmark, the concentrations of various PAHs congeners were converted into BaP equivalent concentrations to obtain TEQ_BaP_, which can be used to reflect the carcinogenic effect of PAHs mixtures [21,26]. The results of Nisbet and Lagoy provided the toxicity equivalent factor (TEF) of each PAHs (BaP was used as a reference, and the toxicity equivalent factor of BaP was regarded as 1 to convert the TEF values of other PAHs) [27,28,29]. The TEQ_BaP_ ingested through seasoning flour products was calculated by the following equation:(3)$$TEQ_{Bap}=\sum \left(C_{j}{\;\times \;\mathrm{TEF}}_{j}\right)$$
where C_j_ is the concentration of individual PAHs congener j (μg/kg); TEF_j_ is the toxicity equivalent factor of PAHs congener j (NAP, ACY, ACE, FLU, PHE, FLA and PYR: 0.001; ANT, CHR, and BghiP: 0.01; BaA, BbF, BkF, and IcdP: 0.1; BaP: 1; DahA: 5) [27].

The estimated exposure of the survey population to PAHs were divided into three groups based on age: children (6–9 years), adolescents (10–19 years), and adults (20–26 years). The daily dietary exposure of TEQ_BaP_ (ED_TEQ_) from seasoning flour products for each group was calculated by the following equation:(4)$$ED_{TEQ}=TEQ_{\mathrm{BaP}}\times IR_{x}$$
where TEQ_Baq_ is the toxic equivalent concentration of seasoning flour products (ng/g); IR_x_ is the daily intake of seasoning flour products for x group (g/d).

The USEPA proposed to use ILCR model to evaluate the carcinogenic risk of PAHs in seasoning flour products [30,31]. The following equation was employed to calculate the ILCR of PAH exposure in each population group:(5)$$\mathrm{ILCR}=\frac{{\mathrm{TEQ}}_{\mathrm{Baq}}{\;\times \;\mathrm{IR}}_{x}{\times \;\mathrm{EF}\;\times \;\mathrm{ED}}_{x}\;\times \mathrm{CSF}\times \mathrm{CF}\;}{\mathrm{BW}\;\times \mathrm{AT}\;}$$
where TEQ_Baq_ is the toxic equivalent concentration of seasoning flour products (ng/g); IR_x_ is the daily ingestion of seasoning flour products per person in group x (g/d); EF is the exposure frequency (365 days/year); ED_x_ is the exposure duration in group x (children: 4 years, adolescents: 10 years, adults: 7 years); CSF is the potential cancer slope factor of BaP (7.3 (mg/kg/d)^−1^) [32]; CF is the conversion factor (10^−6^ mg/ng); BW is the body weight (kg); AT is the average of life span (25,550 days) [33].

### 2.6. Data Analysis

SPSS 25.0 and GraphPad 8.0 were used for data processing and analysis. The average, maximum, P95, and standard difference were used for the statistical description of the data. Normality tests and homogeneity of variance tests were used to determine subsequent statistical analysis methods, either parametric or non-parametric tests. When the data were normally distributed and the variances got homogeneous, statistical significance was calculated by one-way ANOVA, followed by the Fischer least-significant difference (LSD) test (if the number of groups is 3) or Tukey’s multiple comparison’s test (if the number of groups is 4). Otherwise, the difference among various seasoning flour products were analyzed by Kruskal-Wallis (K-W) test. Statistical significance was viewed as *p*-values < 0.05.

## 3. Results

### 3.1. Contamination of PAHs in Seasoning Flour Products Widely Circulated in Hunan Province

The concentrations of total PAHs in the seasoning flour products widely circulated in Hunan Province were analyzed and shown in Table 1. In this study, 79 samples of seasoning flour products were tested, of which 73 samples were detected with PAHs, with a detection rate of 92.41%. According to EFSA recommendation, the lower (LB) and upper (UB) bounds were used when calculating descriptive statistics, that is for values below the LOD, a value of zero or the numerical value of the LOD was entered, respectively. The P_25_, mean, median, P_75_, P_95_, and maximum values of total PAHs were 4.120–7.369, 38.998–40.856, 30.108–30.348, 58.612–59.671, 98.330–98.891, and 165.899–166.759 μg/kg, respectively. According to the taste classification, the detection rate of PAHs in crisp samples and chewy samples reached 100%, the means were 54.988–56.067 and 59.896–61.081 μg/kg, the medians were 54.988–56.067 and 55.635–61.081 μg/kg, and the maximum values were 96.937–96.937 μg/kg and 120.448–121.238 μg/kg, respectively; the detection rate in soggy samples was 88.24%, the mean value was 28.679–30.932 μg/kg, the median were 16.520–29.456 μg/kg, and the maximum value was 165.899–166.759 μg/kg. According to the shape classification, the detection rate of PAHs in rod-shaped samples and flaky samples were both 100%, the mean concentrations were 47.688–49.297 and 49.525–50.975 μg/kg, the median values were 39.171- 39.631 and 53.283–54.684 μg/kg, and the maximum values were 165.899–166.759 and 98.104–98.664 μg/kg; the detection rate of filamentous samples was 81.82%, with a mean value of 9.852–12.711 μg/kg, a median value 4.120–7.380 μg/kg, and a maximum value of 55.975 μg/kg; the detection rate of granular samples was 50%, with a mean of 8.157–11.311 μg/kg, a median of 0.216–3.817 μg/kg, and a maximum value of 47.736–49.516 μg/kg.

Levels of the individual 16 PAHs congeners in the seasoning flour products sampled from Hunan Province are summarized in Table 2. The results showed that FLA had the highest median value (2.769 μg/kg), and BaA had the highest detection rate (81.01%), among which, the median values of BaP was 0.798 μg/kg, which was lower than the Chinese allowable maximum level (Cereals and cereal-based foods: BaP ≤ 5 μg/kg) [34] and the EU allowable maximum level (Processed cereal-based foods: BaP ≤ 1 μg/kg) [16]. From the above data, it appeared that due to the high sensitivity of the analytical methods, despite the relatively high proportion of samples below the LOD, there was a very limited impact of using the LB or UB, which is consistent with the scientific opinion of the Panel on Contaminants in the Food Chain [15]. Therefore, the follow-up results are presented and discussed according to the UB approach. The result of each sample according to the LB approach is available in Appendix A. From the perspective of taste classification, the PAHs with the highest concentration were BaA (1.599 μg/kg, soggy samples), ACE (4.099 μg/kg, crisp samples), and PHE (13.062 μg/kg, chewy samples), and the PAHs with the highest detection rates were BaA (72.55%, 91.67% and 100% in soggy, crisp, and chewy samples, respectively); among the crisp samples, FLA, BaA, CHR, and BaP were all 91.67%. The contents of BaP in the three types of seasoning flour products: soggy samples, crispy samples and chewy samples were 0.348, 1.534, and 1.209 μg/kg, which were lower than the Chinese allowable maximum level. But according to EU standards, the BaP contents in the latter two were excessive (Table 3). Among the different shapes of seasoning flour products, the PAHs with the highest median value in filamentous samples, rod-shaped samples, flaky samples, and granular samples were ANT (0.550 μg/kg), PHE (3.900 μg/kg), FLA (5.246 μg/kg), and ANT (0.550 μg/kg), respectively; the PAHs with higher detection rates were BaA and CHR (50–100%). The concentrations of BaP in the four types of seasoning flour products were 0.320, 0.749, 1.179, and 0.320 μg/kg, which were all in line with the Chinese standard, whereas the flaky samples exceeded the EU standard (Table 4).

The quantitative results of PAHs pollution in various seasoning flour products were described using four alternative indicators (BaP, PAH2, PAH4, and PAH8) recommended by the USEPA. The analysis showed that the median values of BaP in soggy, chewy, and crisp samples were 0.348, 1.209, and 1.534 μg/kg, and the medians of PAH4 were 3.597, 10.212, and 11.245 μg/kg, respectively (Table 3). Among the seasoning flour products classified by shape, the concentrations of PAHs from high to low were in the order of flaky samples, rod-shaped samples, filamentous samples, and granular samples, the corresponding medians of BaP were 1.179, 0.749, 0.320, and 0.320 μg/kg, and the median values of PAH4 were 10.482, 7.629, 1.818, and 0.690 μg/kg, respectively (Table 5). In general, the concentrations of BaP and PAH4 in all kinds of seasoning flour products were lower than the Chinese allowable maximum level (BaP ≤ 5 μg/kg; there is no limit standard for PAH4) [34,35], but the BaP of flaky samples and the PAH4 of flaky, rod-shaped, and filamentous samples were higher than the EU maximum allowable lever (BaP ≤ 1 μg/kg; PAH4 ≤ 1 μg/kg) [16]

### 3.2. Differences in PAHs Pollution in Various Seasoning Flour Products

Since different processing techniques may affect the content of contaminants in food [36,37], the content difference of PAHs in various seasoning flour products were analyzed. The results are shown in Figure 1.

From the perspective of taste types, the total PAHs and PAH4 contents in various samples were ranked from high to low: soggy samples, crisp samples, chewy samples; the BaP contents from high to low were: soggy samples, chewy samples, crisp samples. Among them, the total PAHs and PAH4 contents of the soggy samples were notably lower than those of the crisp samples and chewy samples, while the BaP content was lower than the crisp samples, and the difference with the chewy samples was not noticeable (Figure 1A–C). In all kinds of seasoning flour products with different shapes, the contents of total PAHs, BaP and PAH4 from high to low were in the order of flaky samples, rod-shaped samples, filamentous samples, and granular samples (Figure 1D–F). Regarding total PAHs and PAH4 contents, rod-shaped and flaky samples were higher than filamentous and granular samples (Figure 1D,F). However, in terms of BaP content, the differences among filamentous and rod-shaped samples were not statistically significant (Figure 1E).

### 3.3. Health Risk Assessment of PAHs

Based on the survey results of the average daily dietary consumption of seasoning flour products of residents in Hunan and the concentration of PAHs in the seasoning flour products mentioned above, the EDI values and MOE values of four types of PAHs indicators from seasoning flour products with different tastes were calculated (Table 6). Generally, the smaller the MOE value represents the higher health risk. When the MOE value is greater than 10,000, the intake of concerned harmful substances has no potential risk for ordinary consumers, and when the MOE is less than or equal to 10,000, it will cause some damage to consumers’ health [38,39]. As we can see, the results showed that the MOE values of BaP, PAH2, PAH4, and PAH8 in various seasoning flour products were much higher than 10,000 at the average exposure level or at the high exposure level, indicating that all kinds of seasoning flour products investigated did not pose potential health risks to Hunan population. To further understand the health risks of seasoning flour products for residents in Hunan Province, we then assessed the health risks of dietary exposure to PAHs in seasoning flour products for different populations. According to the age classification criteria of United Nations World Health Organization, participants were divided into three groups: children (6–9 years), adolescents (10–19 years), and adults (20–26 years) [40,41,42,43]. Analysis of dietary data showed that the daily intake of seasoning flour products were significantly higher in adults group (40.527 g/d) and adolescents group (36.989 g/d) than in children group (9.987 g/d) (Figure 2A). Further analysis showed that MOE values of dietary exposure to seasoning flour products in adult group and adolescent group were lower than that of children group, but higher than the warning line of MOE value. This suggests that the PAHs dietary exposure in seasoning flour products had no notable health risk for residents of all ages in Hunan Province.

Since the carcinogenic mechanisms of various PAHs are similar, and BaP is considered to be one of the most carcinogenic PAHs; the equivalent concentration of BaP (TEQ_BaP_) is usually used to reflect the carcinogenic potency of the PAHs mixture [26,31]. According to the concentration data (Table 3) and the TEF values of each PAH component (provided by Nisbet et al.) [27], the TEQ_BaP_ values of total PAHs in soggy samples, crisp samples, and chewy samples were 2.303, 5.884, and 4.284 ng/g, respectively (Appendix A). The overall TEQ_BaP_ value of PAHs in seasoning flour products was 3.082 ng/g (Appendix A).

The cumulative daily dietary exposure to PAHs for each population group was calculated using Equation (4). The ED_TEQ_ values were estimated to follow an increasing order of: children (22.995 ng/d), adolescents (85.169 ng/d), and adults (93.315 ng/d) for the soggy samples, and 58.763, 217.642, and 93.315 ng/d for the above groups of crisp samples. The TEQ_Bap_ daily dietary exposure for the chewy samples was observed to be 42.779, 158.442, and 173.598 ng/d for children, adolescents, and adults, respectively. In terms of the overall level of seasoning flour products, the ED_TEQ_ values in children, adolescents, and adults were 30.783, 114.012, and 124.917 ng/d, respectively (Figure 2B). According to the seasoning flour products categories, crisp samples contributed the most to PAHs contamination, followed by soggy samples and chewy samples. Among all population groups, the exposure of adult and adolescent groups was significantly higher than that of children (Figure 2B).

Subsequently, ILCR was used to assess the carcinogenic risk of PAHs in seasoning flour products to different populations. According to the USEPA, an ILCR of 10^−6^ or below indicates a risk level that is considered negligible; an ILCR between 10^−6^ and 10^−4^ indicates a potential cancer risk; an ILCR of 10^−4^ or above indicates a risk that requires high attention [1,44,45,46]. The calculated ILCR values of children, adolescents, and adults were 5.52 × 10^−7^, 2.522 × 10^−6^, and 1.648 × 10^−6^, respectively, which were all lower than the warning value of ILCR (10^−4^). The ILCR value of PAHs for children was less than 10^−6^, which could be considered as a negligible carcinogenic risk; the ILCR values of PAHs for adolescents and adults were higher than 10^−6^ but lower than 10^−5^, indicating a potentially low health risk.

## 4. Discussion

In recent years, food safety issues have received increasing attention. As a kind of convenient leisure food, seasoning flour products are popular among young people, and have an annual output value of nearly 10 billion dollars [10,11]. However, due to the low technical requirements of production and processing and the lack of corresponding management standards, seasoning flour products always have food safety problems [9]. Because of the high temperature frying processing technology, PAHs, which have teratogenic, carcinogenic, and mutagenic toxicity, have become one of the important pollutants of seasoning flour products [12]. PAHs exposure to human beings is strongly associated with the occurrence of several cancers [47,48,49], and were enlisted as priority pollutants by the USEPA [1]. Therefore, as seasoning flour products become an increasingly popular snack among young people worldwide, it is necessary to strengthen the supervision and detection of PAHs in seasoning flour products.

This study assessed the concentrations of 16 PAHs in seasoning flour products sampled from Hunan Province. The results showed that PAHs contamination was common in seasoning flour products, which was in accord with Sha et al.’s result [14]. PAHs were detected in 76 samples of 79 seasoning flour products (Table 1). Among all analyzed samples, the PAHs pollutants with higher detected amounts were BaA, PHE, CHR, and FLA, while the PAHs with the highest detection rate was BaA (Table 2). The concentrations of BaP (0.798 μg/kg) and PAH4 (7.616 μg/kg) detected in this study were higher than those of Sha et al. (Table 5) [14]. The contents of BaP in various seasoning flour products met the Chinese food standard (BaP ≤ 5 μg/kg). But according to the European Union standard, the concentration of BaP and PAH4 in a high proportion of seasoning flour products exceeded the maximum allowable concentration (BaP ≤ 1 μg/kg; PAH4 ≤ 1 μg/kg). There were significant differences (*p* < 0.05) in PAHs contamination of seasoning flour products with different taste classifications (soggy, crisp, chewy) or shape classifications (filamentous, rod-shaped, flaky, granular). Among the seasoning flour product samples with different taste categories, the contents of the 16 PAHs, BaP, and PAH4 in soggy samples were significantly lower than those in crisp samples and chewy samples (Figure 1A–C). Among the seasoning flour products classified by different shapes, the contents of the 16 PAHs, BaP, and PAH4 in the rod-shaped samples and flaky samples were significantly higher than those in the filamentous samples and granular samples (Figure 1G–I).

Based on the consumption and the PAHs contamination of seasoning flour products, the health risks of PAHs dietary exposure in Hunan population were evaluated. According to the recommendations of the Joint FAO/WHO Expert Committee Food Additives (JECFA), USEPA, and EFSA, the MOE value and ILCR value were used to assessment the risk of human health for PAHs exposure [25,50]. On the one hand, the MOE values of BaP, PAH2, PAH4, and PAH8 of various seasoning flour products were much greater than 10,000 at the average and high exposure levels (Table 6). On the other hand, the MOE values and ILCR values of all age groups population (children, adolescents, and adults) did not reach the warning value (MOE < 10,000; ILCR > 10^−4^) (Table 6, Figure 2C). In all age categories, adolescents showed the highest carcinogenic risk (2.522 × 10^−6^), followed by adults (1.648 × 10^−6^) and children (5.52 × 10^−7^) (Figure 2C). Interestingly, Aamir et al. showed that in the Pakistani population, adults have the highest carcinogenic risk of exposure to dietary PAHs, followed by children, the elderly, and adolescents [1]. The findings of Ju et al. suggest that the health risks of seniors and children are higher than those of adolescents and adults [6]. We hypothesized that the difference was due to the ingestion amount. The food investigated in this study, seasoning flour products, is a kind of spicy, high-oil, and high-salt food. Children’s intake of seasoning flour products is severely restricted by their parents, so it is significantly lower than the rest of the groups. The adolescent group has the highest ILCR value, which could be attributed to the light body-weight, the higher intake, and the relatively short exposure duration. This inference is consistent with many previous studies [51,52]. Overall, based on the calculated MOE and ILCR values, the dietary exposure of PAHs in seasoning flour products circulating in Hunan area has a relatively low health risk to Hunan population. Despite the health risks were tolerable, the co-exposure and long-term exposure to PAHs might produce higher health risks than those commented here. Therefore, the susceptible populations, such as children and adolescents, are recommended to limit the intake of seasoning flour products.

This pilot study is beneficial to understand the contamination of PAHs in various seasoning flour products and the potential health hazards to people. It is expected to provide consumers and government departments with health-related information on seasoning flour products, which will further help consumers choose a healthy diet and provide a reference for establishing and improving PAHs limit standards in seasoning flour products. However, due to the significant influence of processing technology and the production environment, the concentration of PAHs in seasoning flour products of different brands and batches may be significantly different [9]. In the future, seasoning flour products circulating in more areas should be collected to assess their health risks to regional populations.

## 5. Conclusions

This study focused on detecting 16 PAHs in seasoning flour products circulating in Hunan Province, China, and the health risk assessment for residents in Hunan Province. This is beneficial to fill the gap in the research on the health risks of dietary exposure of PAHs in seasoning flour products. The results showed that PAHs contamination was common in seasoning flour products. Among them, the concentrations of BaP and PAH4 were in line with the Chinese standard, but higher than the maximum allowable lever of the European Union. It is worth noting that the contents of PAHs in seasoning flour products with different taste or shape were significantly different. The contents of PAHs in soggy samples were significantly lower than those in crisp and chewy samples, and the PAHs content in rod-shaped and flaky samples were significantly higher than those in filamentous and granular samples. We suggest that consumers should be inclined to choose soggy, filamentous, or granular seasoning flour products when purchasing, and it is necessary to control the intake. Although the current health risk assessment results showed that PAHs in seasoning flour products have a low health risk for the Hunan population, we recommend that the corresponding detection and health risk assessment should be carried out in a wider area, and the control standards of PAHs contamination in seasoning flour products should be further improved.

## Figures and Tables

**Figure 1 ijerph-20-00963-f001:**
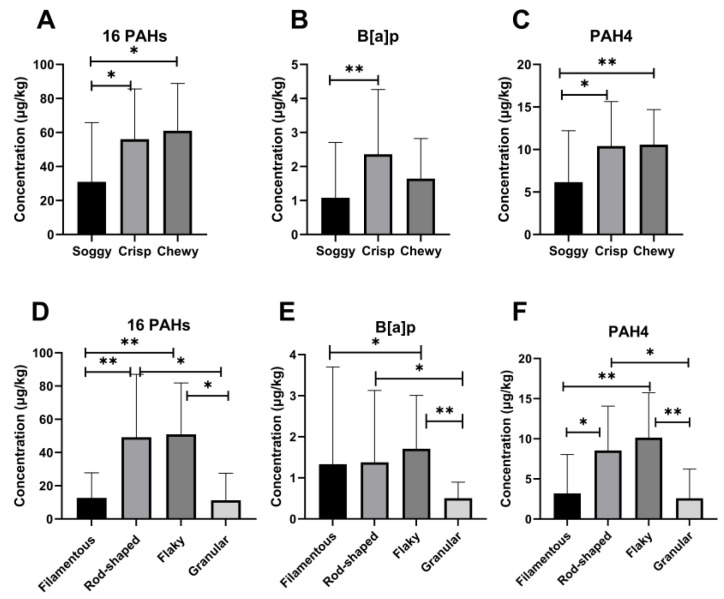
The difference analysis of PAHs content in various seasoning flour products. (**A**–**C**) The concentration of 16 PAHs (**A**), BaP (**B**), and PAH4 (**C**) in the seasoning flour products with different tastes. (**D**–**F**) The concentration of 16 PAHs (**D**), BaP (**E**), and PAH4 (**F**) in seasoning flour products of different shapes. * means *p* < 0.05; ** means *p* < 0.01.

**Figure 2 ijerph-20-00963-f002:**
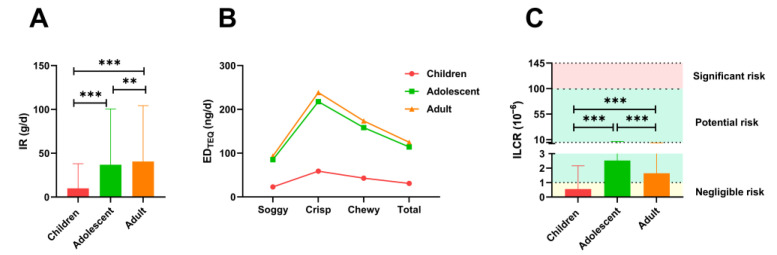
Incremental lifetime cancer risk of PAHs in seasoning flour products for various populations. (**A**) The daily intake (IR) of seasoning flour products for each population group. (**B**) The daily dietary exposure of toxicity equivalent concentration from seasoning flour products for each population group. (**C**) Incremental lifetime cancer risk (ILCR) values of each population group. ** means *p* < 0.01; *** means *p* < 0.001.

**Table 1 ijerph-20-00963-t001:** The concentrations of total PAHs in the seasoning flour products widely circulated in Hunan Province.

Seasoning Flour Product	>LOD (%)	P_25_ (μg/kg)		Mean (μg/kg)		Median (μg/kg)		P_75_ (μg/kg)		P_95_ (μg/kg)		Maximum (μg/kg)	
		LB	UB	LB	UB	LB	UB	LB	UB	LB	UB	LB	UB
Taste													
Soggy (*n* = 51)	88.24	2.102	5.074	28.679	30.932	16.520	29.456	49.634	50.084	97.670	98.475	165.899	166.759
Crisp (*n* = 12)	100	22.768	23.727	54.988	56.067	51.836	52.876	86.394	88.149	93.861	93.982	96.937	96.937
Chewy (*n* = 16)	100	46.636	47.036	59.896	61.081	55.635	56.720	78.294	79.674	105.383	106.143	120.448	121.238
Shape													
Filamentous (*n* = 11)	81.82	2.102	5.032	9.852	12.711	4.120	7.380	9.691	11.931	35.872	37.512	55.975	55.975
Rod-shaped (*n* = 35)	100.00	21.368	23.188	47.688	49.227	39.171	39.631	63.889	65.409	107.447	108.461	165.899	166.759
Flaky (*n* = 25)	100.00	18.344	20.899	49.525	50.975	53.283	54.684	76.193	76.843	94.989	95.033	98.104	98.664
Granular (*n* = 8)	50.00	0.000	3.640	8.157	11.311	0.216	3.817	12.197	14.524	36.575	38.429	47.736	49.516
Total (*n* = 79)	92.41	4.120	7.369	38.998	40.856	30.108	30.348	58.612	59.671	98.330	98.891	165.899	166.759

>LOD (%): the proportion of content above the limit of detection. LB: the lower bound (all samples below LOD or LOQ replaced by 0). UB: the upper bound (all samples below LOD or LOQ replaced by the level of detection or the lever of quantification).

**Table 2 ijerph-20-00963-t002:** Contamination of 16 PAHs congeners in seasoning flour products.

PAH	Total (*n* = 79)										
	>LOD (%)	P_25_ (μg/kg)		Mean (μg/kg)		Median (μg/kg)		P_75_ (μg/kg)		Maximum (μg/kg)	
		LB	UB	LB	UB	LB	UB	LB	UB	LB	UB
NAP	55.70	0.000	0.160	5.232	5.303	0.524	0.524	3.799	3.799	77.677	77.677
ACY	41.77	0.000	0.320	1.825	2.011	0.000	0.320	2.091	2.091	24.093	24.093
ACE	51.90	0.000	0.280	1.603	1.737	0.346	0.346	3.072	3.072	9.835	9.835
FLU	62.03	0.000	0.350	3.199	3.331	1.539	1.539	5.751	5.751	20.167	20.167
PHE	65.82	0.000	0.040	8.055	8.069	2.047	2.047	13.572	13.572	58.961	58.961
ANT	40.51	0.000	0.550	1.121	1.448	0.000	0.550	1.774	1.774	6.675	6.675
FLA	67.09	0.000	0.040	5.315	5.328	2.769	2.769	7.948	7.948	26.470	26.470
PYR	64.56	0.000	0.120	2.778	2.821	1.896	1.896	4.357	4.357	13.158	13.158
BaA	81.01	0.850	0.850	2.762	2.777	2.480	2.480	3.916	3.916	10.178	10.178
CHR	75.95	0.318	0.318	2.836	2.856	2.447	2.447	4.590	4.590	10.600	10.600
BbF	29.11	0.000	0.210	0.528	0.677	0.000	0.000	0.608	0.608	4.387	4.387
BkF	44.30	0.000	0.300	1.312	1.479	0.000	0.300	2.530	2.530	6.484	6.484
BaP	65.82	0.000	0.320	1.279	1.389	0.798	0.798	1.689	1.689	8.184	8.184
IcdP	53.16	0.000	0.330	0.546	0.701	0.379	0.379	0.677	0.677	3.997	3.997
DahA	18.99	0.000	0.240	0.350	0.545	0.000	0.240	0.000	0.240	4.243	4.243
BghiP	41.77	0.000	0.220	0.256	0.384	0.000	0.200	0.446	0.446	2.272	2.272

NAP, naphthalene; ACY, acenaphthylene; ACE, acenaphthene; FLU, fluorene; PHE, anthracene; FLA, fluoranthene; Pyr, pyrene; BaA, benzo[a]anthracene; CHR, chrysene, BbF, benzo[b]fluoranthene; BkF, benzo[k]fluoranthene; BaP, benzo[a]pyrene; IcdP, indene[1,2,3-c,d]pyrene; DahA, dibenz[a,h]anthracene; BghiP, benzo[g,h,i]perylene.

**Table 3 ijerph-20-00963-t003:** Contamination of 16 PAHs congeners in seasoning flour products with different tastes.

PAH	Taste					
	Soggy (*n* = 51)		Crisp (*n* = 12)		Chewy (*n* = 16)	
	>LOD (%)	Median (P_25_, P_75_) (μg/kg)	>LOD (%)	Median (P_25_, P_75_) (μg/kg)	>LOD (%)	Median (P_25_, P_75_) (μg/kg)
		UB		UB		UB
NAP	45.10	0.160 (0.160, 3.076)	66.67	2.992 (0.160, 6.603)	81.25	1.667 (0.289, 15.346)
ACY	29.41	0.320 (0.320, 0.742)	58.33	3.180 (0.320, 5.626)	68.75	0.798 (0.320, 2.404)
ACE	31.37	0.280 (0.280, 0.668)	83.33	4.099 (0.831, 7.558)	93.75	1.433 (0.568, 3.727)
FLU	49.02	0.350 (0.350, 3.570)	83.33	1.792 (0.809, 3.566)	87.50	6.178 (0.633, 8.909)
PHE	52.94	0.797 (0.040, 9.082)	83.33	2.197 (0.519, 3.988)	93.75	13.062 (2.088, 29.181)
ANT	37.25	0.550 (0.550, 1.261)	58.33	1.704 (0.550, 4.431)	37.50	0.550 (0.550, 2.709)
FLA	52.94	0.954 (0.040, 7.395)	91.67	3.819 (0.600, 5.332)	93.75	8.662 (2.059, 15.340)
PYR	50.98	0.307 (0.120, 2.582)	83.33	3.152 (0.614, 4.730)	93.75	4.483 (2.525, 7.559)
BaA	72.55	1.599 (0.080, 3.572)	91.67	2.889 (1.980, 3.783)	100.00	4.464 (1.779, 6.006)
CHR	66.67	1.107 (0.080, 3.865)	91.67	3.304 (2.2344.485)	93.75	4.182 (1.913, 6.552)
BbF	21.57	0.210 (0.210, 0.210)	50.00	0.378 (0.210, 3.256)	37.50	0.210 (0.210, 1.785)
BkF	33.33	0.300 (0.300, 2.157)	83.33	2.776 (2.029, 3.817)	50.00	0.328 (0.300, 2.424)
BaP	50.98	0.348 (0.320, 1.028)	91.67	1.534 (0.770, 4.024)	93.75	1.209 (0.902, 2.274)
IcdP	37.25	0.330 (0.330, 0.463)	75.00	0.696 (0.366, 2.381)	87.50	0.482 (0.353, 0.838)
DahA	11.76	0.240 (0.240, 0.240)	41.67	0.240 (0.240, 3.033)	25.00	0.240 (0.240, 0.326)
BghiP	37.25	0.220 (0.220, 0.322)	50.00	0.386 (0.220, 0.714)	50.00	0.288 (0.220, 0.498)

**Table 4 ijerph-20-00963-t004:** Contamination of 16 PAHs congeners in seasoning flour products with different shapes.

PAH	Shape							
	Filamentous (*n* = 11)		Rod-shaped (*n* = 35)		Flaky (*n* = 25)		Granular (*n* = 8)	
	>LOD (%)	Median (P25, P75) (μg/kg)	>LOD (%)	Median (P25, P75) (μg/kg)	>LOD (%)	Median (P25, P75) (μg/kg)	>LOD (%)	Median (P25, P75) (μg/kg)
		UB		UB		UB		UB
NAP	18.18	0.160 (0.160, 0.160)	77.14	2.917 (0.848, 13.208)	56.00	0.426 (0.160, 3.939)	12.50	0.160 (0.160, 0.160)
ACY	27.27	0.320 (0.320, 0.782)	48.57	0.320 (0.320, 2.633)	52.00	0.360 (0.320, 2.716)	0.00	0.320 (0.320, 0.320)
ACE	9.09	0.280 (0.280, 0.280)	62.86	0.853 (0.280, 3.203)	68.00	0.623 (0.280, 4.681)	12.50	0.280 (0.280, 0.280)
FLU	9.09	0.350 (0.350, 0.350)	80.00	2.201 (0.601, 6.798)	76.00	2.363 (0.471, 6.729)	12.50	0.350 (0.350, 0.350)
PHE	18.18	0.040 (0.040, 0.040)	82.86	3.900 (0.971, 14.521)	80.00	3.779 (0.560, 19.774)	12.50	0.040 (0.040, 0.040)
ANT	36.36	0.550 (0.550, 1.774)	48.57	0.550 (0.550, 1.952)	44.00	0.550 (0.550, 2.429)	0.00	0.550 (0.550, 0.550)
FLA	27.27	0.040 (0.040, 0.954)	80.00	5.135 (0.171, 7.920)	80.00	5.246 (1.578, 13.140)	25.00	0.040 (0.040, 3.066)
PYR	9.09	0.120 (0.120, 0.120)	77.14	2.154 (0.186, 4.102)	84.00	3.836 (1.607, 6.641)	25.00	0.120 (0.120, 0.830)
BaA	54.55	0.278 (0.080, 1.333)	91.43	3.006 (1.711, 3.916)	92.00	3.572 (1.856, 5.674)	37.50	0.080 (0.080, 2.164)
CHR	36.36	0.080 (0.080, 0.802)	91.43	3.289 (1.773, 4.612)	88.00	3.538 (2.047, 5.768)	25.00	0.080 (0.080,1.582)
BbF	18.18	0.210 (0.210, 0.210)	34.29	0.210 (0.210, 0.703)	36.00	0.210 (0.210, 1.420)	0.00	0.210 (0.210, 0.210)
BkF	36.36	0.300 (0.300, 2.854)	42.86	0.300 (0.300, 2.480)	56.00	1.357 (0.300, 3.399)	25.00	0.300 (0.300, 1.007)
BaP	27.27	0.320 (0.320, 1.953)	77.14	0.749 (0.348, 1.307)	80.00	1.179 (0.713, 2.763)	25.00	0.320 (0.320, 0.609)
IcdP	9.09	0.330 (0.330, 0.330)	60.00	0.400 (0.330, 0.575)	72.00	0.552 (0.330, 1.091)	25.00	0.330 (0.330, 0.429)
DahA	9.09	0.240 (0.240, 0.240)	22.86	0.240 (0.240, 0.240)	24.00	0.240 (0.240, 0.332)	0.00	0.240 (0.240, 0.240)
BghiP	9.09	0.220 (0.220, 0.220)	42.86	0.220 (0.220, 0.446)	60.00	0.299 (0.220,0.554)	25.00	0.220 (0.220, 0.283)

**Table 5 ijerph-20-00963-t005:** Concentrations of four PAHs pollution indicators in various seasoning flour products.

Seasoning Flour Product	BaP		PAH2		PAH4		PAH8	
	Median (P25, P75) (μg/kg)	P_95_ (μg/kg)	Median (P25, P75) (μg/kg)	P_95_ (μg/kg)	Median (P25, P75) (μg/kg)	P_95_ (μg/kg)	Median (P25, P75) (μg/kg)	P_95_ (μg/kg)
Taste								
Soggy (*n* = 51)	0.348 (0.320, 1.028)	3.911	2.033 (0.400, 4.831)	12.347	3.597 (0.888, 8.957)	18.364	5.513 (2.937, 11.357)	24.862
Crisp (*n* = 12)	1.534 (0.770, 4.024)	5.263	6.032 (3.008, 8.238)	9.735	11.245 (5.399, 15.828)	16.122	15.999 (8.375, 22.471)	28.860
Chewy (*n* = 16)	1.209 (0.902, 2.274)	3.458	5.411 (4.218, 7.576)	8.720	10.212 (7.664, 13.789)	16.147	14.055 (10.895, 16.651)	19.411
Shape								
Filamentous (*n* = 11)	0.320 (0.320, 1.953)	5.077	0.659 (0.400, 2.033)	7.617	1.818 (0.690, 2.665)	10.569	3.755 (2.908, 4.687)	18.392
Rod-shaped (*n* = 35)	0.749 (0.348, 1.307)	5.306	4.094 (2.302, 6.704)	11.285	7.629 (4.423, 11.764)	18.305	9.445 (5.903, 14.518)	26.488
Flaky (*n* = 25)	1.179 (0.713, 2.763)	4.077	6.477 (3.986, 7.284)	8.751	10.482 (7.585, 13.738)	16.007	14.323 (9.175, 18.595)	25.016
Granular (*n* = 8)	0.320 (0.320, 0.609)	1.165	0.400 (0.400, 2.190)	4.700	0.690 (0.690, 4.5650	8.819	1.957 (1.780, 8.821)	12.072
Total (*n* = 79)	0.798 (0.320, 1.689)	4.528	3.998 (0.662, 6.704)	11.008	7.616 (2.124, 12.513)	17.579	9.445 (3.755, 14.733)	26.000

**Table 6 ijerph-20-00963-t006:** Daily dietary exposure and the margin of exposure of PAHs.

	Pollutant	BMDL_10_ [mg/(kg·BW)/d]	EDI		MOE	
			P_50_ [ng/(kg·BW)/d]	P_95_ [ng/(kg·BW)/d]	P_50_	P_95_
Seasoning flour product						
Soggy	BaP	0.07	0.133	1.498	525,194	46,732
	PAH2	0.17	0.779	4.729	218,330	35,949
	PAH4	0.34	1.378	7.033	246,797	48,341
	PAH8	0.49	2.111	9.522	232,065	51,459
Crisp	BaP	0.07	0.245	0.842	285,202	83,127
	PAH2	0.17	0.965	1.558	176,144	109,142
	PAH4	0.34	1.799	2.580	188,973	131,807
	PAH8	0.49	2.560	4.618	191,418	106,116
Chewy	BaP	0.07	0.214	0.612	327,114	114,367
	PAH2	0.17	0.958	1.543	177,500	110,144
	PAH4	0.34	1.808	2.858	188,103	118,964
	PAH8	0.49	2.488	3.436	196,966	142,618
Population group						
Children	BaP	0.07	0.342	1.943	204,474	36,036
	PAH2	0.17	1.715	4.722	99,117	35,998
	PAH4	0.34	3.267	7.541	104,063	45,085
	PAH8	0.49	4.052	11.154	120,931	43,930
Adolescent	BaP	0.07	0.626	3.554	111,744	19,693
	PAH2	0.17	3.138	8.641	54,167	19,673
	PAH4	0.34	5.979	13.800	56,870	24,639
	PAH8	0.49	7.414	20.410	66,088	24,008
Adult	BaP	0.07	0.585	3.319	119,672	21,091
	PAH2	0.17	2.931	8.069	58,010	21,069
	PAH4	0.34	5.583	12.885	60,904	26,386
	PAH8	0.49	6.923	19.058	70,777	25,711

EDI, estimated daily intake; BMDL_10_, benchmark dose lower confidence limit; MOE, margin of exposure. The BMDL_10_ values were obtained from EFSA.

## Data Availability

The data that support the findings of this study are not publicly available due to the data containing information that could compromise participants privacy but are available from the corresponding author on reasonable request.

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
