# Peer review of "Contamination and Health Risk Assessment of Polycyclic Aromatic Hydrocarbons in Seasoning Flour Products in Hunan, China"

_ijerph, 2023, doi:10.3390/ijerph20020963_

Round 1
Reviewer 1 Report
Considering that the manuscript's objectives were to conduct an exposure assessment, which is a very complex and serious issue, my opinion is as follows.
In order to obtain reliable and robust results for exposure assessment, particularly for cancerogenic compounds, the number of analyzed samples, the period of investigation, and the origin of collected samples must be sufficient and systematic, as a whole to represent a realistic scenario, draw conclusions and propose further actions which will be in function of food safety improvement.
Otherwise, results can be unreliable. Thus, this is the main shortcoming of this research. The number of analyzed samples must cover different periods of the year. Samples must be taken from different regions that further correspond with sources of PAH.
Also, the processing of results and presentation of obtained results are not in accordance with the ones proposed by the international agencies in terms of exposure assessment.
My suggestion is inserted in the comments .

Author Response
Response to Reviewer 1
Dear Reviewer,
We sincerely appreciate your insightful and valuable comments and suggestions. All of your comments and suggestions are of great help to the improvement of the quality of our manuscript. Here we submit a new version of our revised manuscript according to the suggestions, titled “Contamination and health risk assessment of polycyclic aromatic hydrocarbons in seasoning flour products in Hunan, China” (ID: ijerph-2126415). You will see the difference made to the revised manuscript.
If you have any questions about this paper, please do not hesitate to contact me.
Sincerely yours,
Hong Qin
Extensive revisions to the manuscript have been marked up using the “Track Changes” function. Please find below an itemized list of POINT-BY-POINT responses to these comments:
Reviewer #1
Comments and Suggestions for Authors
Considering that the manuscript's objectives were to conduct an exposure assessment, which is a very complex and serious issue, my opinion is as follows.
In order to obtain reliable and robust results for exposure assessment, particularly for cancerogenic compounds, the number of analyzed samples, the period of investigation, and the origin of collected samples must be sufficient and systematic, as a whole to represent a realistic scenario, draw conclusions and propose further actions which will be in function of food safety improvement.
Otherwise, results can be unreliable. Thus, this is the main shortcoming of this research. The number of analyzed samples must cover different periods of the year. Samples must be taken from different regions that further correspond with sources of PAH.
Rev: Thanks for your insightful comments. We understand that sufficient and systematic samples are important for exposure assessment of PAHs. But the research on PAHs in seasoning flour products is very limited, so we cannot obtain data related to PAHs pollution in seasoning flour products in different periods and regions, which is a shortcoming of this study. However, in order to reflect the pollution situation of seasoning flour products as reliable as possible, we chose Hunan Province, a major producing area of seasoning flour products, as the investigation site. In the sampling process, a various potential influencing factors, such as manufacturer, sales location, brand, packaging type, shape and taste, were taken into account. The samples covered all kinds of seasoning flour products in local circulation. And the number of analyzed samples, the period of investigation, and the origin of collected samples have been described in the manuscript. Moreover, we will take your suggestions into full consideration in the follow-up study, and collect samples in different periods and more origins.
Also, the processing of results and presentation of obtained results are not in accordance with the ones proposed by the international agencies in terms of exposure assessment.
Rev: Thank you for your valuable suggestion. Accordingly, we have reprocessed the data according to the model of international agencies, and adjusted the presentation of all the results in the revised manuscript.
My suggestion is inserted in the comments.
Rev: Thanks for your meticulous review and all the valuable comments. To facilitate reading, we have transcribed all the comments from the peer-review-25582250.v1.pdf and replied point-to-point as below.
Line33: Legislative framework for setting maximum levels for certain contaminants in foodstuffs and proposed maximum daily intake is not considered. Thus, this section must be improved.
Rev: Thank you for your suggestion. We have made adjustments according to your advice. In our revision, we have added the consideration on maximum daily intake and maximum level of PAHs in seasoning flour products (Line 67-71 in the revised manuscript).
Line170: Despite different units being used, I suppose that the calculation was properly conducted.
Rev: We checked the calculation process again to make sure the results were correct.
Line205: table 1: Descriptive statistics for the concentration in μg/kg of total PAHs in the seasoning flour products is not properly presented. Considering the huge differences between the mean level and maximum level descriptive statistics must comprises >LOD (%), distribution of concentration p25, mean, median, p75, p95 (percentile), maximum.
This is very important for exposure assessment following dietary intake of PAHs. In addition, according to EFSA recommendation, the lower (LB) and upper (UB) bounds have to be used when calculating descriptive statistics, that is for values below the LOD, for all of the calculated values.
Rev: Thanks for sharing this important advice. According to your recommendation, we have converted the values below the LOD to zero (LB) and the numerical value of the LOD (UB) [1]. And the >LOD (%), P25, mean, median, P75, P95 and maximum were used for descriptive statistics to better describe the real situation of the data (Line 258-281 in the revised manuscript).
[1] EFSA (European Food Safety Authority), 2008. Polycyclic aromatic hydrocarbons in food Scientific Opinion of The Panel on Contaminants in The Food: EFSA J. 724, 1–114.
Line229: table 2: As before regarding to descriptive statistics data. Divide table 2 into two separate tables which are Taste data and Shape data.
Rev: Thanks for your suggestion. In order to show all the data more clearly, the Table 2 have been divided into three separate tables, which are Total data, Taste data and Shape data. And the >LOD (%), percentiles (P25, P50, P75), mean and maximum were used to describe the contents of various PAHs in seasoning flour products under LB and UB conditions (Line 309-347 in the revised manuscript).
After reviewing relevant literatures, we found that due to the high sensitivity of the analytical methods, despite the relatively high proportion of samples below the LOD, there was a very limited impact of using the LB or UB [1,2]. And the data of the Table 1 and the Total data of Table 2 supported this opinion. Therefore, the follow-up descriptive statistics data were presented and discussed according to the UB approach. And the result of each sample according to the LB approach is available in Table S1-S6.
[1] EFSA (European Food Safety Authority), 2008. Polycyclic aromatic hydrocarbons in food Scientific Opinion of The Panel on Contaminants in The Food: EFSA J. 724, 1–114.
[2] Schendel S, Berg T, Scherfling M, et al. Results of the BfR MEAL Study: Highest levels of retinol found in animal livers and of β-carotene in yellow-orange and green leafy vegetables. Food Chem X. 2022 Sep 26;16:100458. doi: 10.1016/j.fochx.2022.100458. PMID: 36203952; PMCID: PMC9530835.
Line247: table3: As before, regarding to descriptive statistics data. Use EFSA model adopted for presenting Descriptive statistics for the concentration (μg/kg) of PAHs in food products (Scientific Opinion of the Panel on Contaminants in the Food Chain on a request from the European Commission on Polycyclic Aromatic Hydrocarbons in Food. The EFSA Journal (2008) 724, 1-114).
Rev: Thanks for your valuable advice. According to your suggestion, we recalculated and analyzed the data according to LB and UB approach, respectively. Moreover, considering the skewed distribution of the data, the median, P25, P75 and P95 were used to describe the data distribution comprehensively (Line 362-373 in the revised manuscript).
Line275: Why did not used JECFA approach to calculate MOE. JECFA proposed mean and high-level intakes of PAHs with the calculated benchmark dose lower confidence limit for PAHs and calculated MOEs of 25,000 and 10,000 respectively. (JECFA (2005). Joint FAO/WHO Expert Committee on Food Additives. Sixty-fourth meeting, Rome, 8-17 February 2005. Summary and Conclusions. Accessed on 19 June 2005 at http://www.who.int/ipcs/food/jecfa/summaries/summary_report_64_final.pdf.)
Based on these MOEs, you can drawn concern for human health.
Rev: Thanks for your advice. The European Food Safety Authority (EFSA) and the World Health Organization (WHO), with the support of the International Life Sciences Institute European Branch (ILSI Europe), organized an international conference on 16- 18 November 2005 to discuss how regulatory and advisory bodies evaluate the potential risks of the presence in food of substances that are both genotoxic and carcinogenic. A 10% increased tumor incidence over background was chosen as the benchmark response (BMR) in the EFSA Opinion and the JECFA Report. The uncertainties that would have to be allowed for before any MOE could be considered to be of low concern from the public health point of view. This issue was considered most comprehensively in the EFSA Opinion, which concluded that an MOE of 10,000 and above, based on a BMDL10 from an animal study, would be a value that would indicate a low concern from a public health point of view and that might be considered a low priority for risk management actions [1]. Moreover, through literature review, we found that a large number of risk assessment studies adopted 10,000 as the risk mark [2-6].
Of course, your suggestion has also been taken into account, using JECFA's criteria for average and high-level exposure of 25,000 and 10,000, respectively. The results are consistent with our current conclusion that PAHs dietary exposure in seasoning flour products had no notable health risk for residents of Hunan Province (Line 399-425 in the revised manuscript).
- Barlow S, Renwick AG, Kleiner J, et al. Risk assessment of substances that are both genotoxic and carcinogenic report of an International Conference organized by EFSA and WHO with support of ILSI Europe. Food Chem Toxicol. 2006 Oct;44(10):1636-50. doi: 10.1016/j.fct.2006.06.020. Epub 2006 Jul 8. PMID: 16891049.
- Ju YR, Chen CF, Wang MH, et al. Assessment of polycyclic aromatic hydrocarbons in seafood collected from coastal aquaculture ponds in Taiwan and human health risk assessment. J Hazard Mater. 2022 Jan 5;421:126708. doi: 10.1016/j.jhazmat.2021.126708. Epub 2021 Jul 21. PMID: 34352521.
- Javed A, Naeem I, Benkerroum N, et al. Occurrence and Health Risk Assessment of Aflatoxins through Intake of Eastern Herbal Medicines Collected from Four Districts of Southern Punjab-Pakistan. Int J Environ Res Public Health. 2021 Sep 10;18(18):9531. doi: 10.3390/ijerph18189531. PMID: 34574455; PMCID: PMC8466447.
- Zhang X, Wang Z, Liu L, et al. Assessment of the risks from dietary lead exposure in China. J Hazard Mater. 2021 Sep 15;418:126134. doi: 10.1016/j.jhazmat.2021.126134. Epub 2021 May 24. PMID: 34119975.
- Meerpoel C, Vidal A, Andjelkovic M, et al. Dietary exposure assessment and risk characterization of citrinin and ochratoxin A in Belgium. Food Chem Toxicol. 2021 Jan;147:111914. doi: 10.1016/j.fct.2020.111914. Epub 2020 Dec 8. PMID: 33307117.
- Guizellini GM, Sampaio GR, da Silva SA, et al. Concentration and potential health risk of polycyclic aromatic hydrocarbons for consumers of chocolate in Brazil. Food Chem. 2023 Mar 30;405(Pt B):134853. doi: 10.1016/j.foodchem.2022.134853. Epub 2022 Nov 7. PMID: 36435108.
Line293: table 4: Considering the huge differences between level of contamination in examined samples, because of the potential for chronic effects of PAHs, the use of concentrations expressed in percentiles would represent in theory the most realistic picture for long term dietary exposure. Thus this section must be improved.
Rev: We sincerely appreciate your insightful suggestions. Considering the huge variation in pollution levels in examined samples and the potential chronic effects of PAHs, long-term dietary exposure was assessed using concentrations represented by P50 and P95 (Line 422-425 in the revised manuscript).

Reviewer 2 Report
The article entitled “Contamination and health risk assessment of polycyclic aromatic hydrocarbons in seasoning flour products in Hunan, China” describes an analytical methodology to analyse 16 priority PAHs in seasoning flour products. In addition, an exhaustive risk evaluation was performed considering the concentrations obtained and the consumption pattern for three ranges of age, children, adolescents and adults.
As authors confirm, there are only 2 articles dedicated to analysing PAHs in this kind of products, but these articles evaluated only a small number of PAHs. The number of analysed samples is big enough to be considered a representative study and the high detection rate (96.20%) highlights the importance of controlling these food samples. In my opinion, the present article has novelty enough to be published in International Journal of Environmental Research and Public Health. However, there are some minor changes that must be addressed before publication:
*General commentary: As the author mentioned, there is scarce information about PAHs in these food products. Thus, the novelty of this work should be highlighted along the manuscript.
*Line 50: I recommend putting a range of age instead put “millennials and post-millennials”
*Line 53: The reference 14 (Investigation and analysis of market latiao food related indexes) could not be found. I recommend to authors to find another reference which support their affirmation
*Line 70-71: The same objective was described in lines 75-79. Merge both sentences
*Line 90: I understand you crush the samples prior to the extraction process. If yes, please specify. On the other hand, what’s the matter to make a salting-out QuEChERS? I believe that samples called “spicy strip” are solid samples with a minimal water content, so you will have only the organic phase, you do not need to separate any phase. Correct me if I am wrong, please
*Line 91: Change “NaCL” for “NaCl”. Besides, it was not previously abbreviated
*Line 95: When you use cold centrifugation specify the temperature used
*Line 98: Eliminate “And”
*Line 100: Replace “2min” for “2 min”
*Line 128: Include the questionnaire questions in supplementary material
*Line 259: You said that for all seasoning products, contain was from high to low in the order of filamentous samples, rod-shaped samples, flaky samples and granular samples. However, in the next lines you said that rod-shaped samples and flaky samples concentrations were higher than filamentous samples for total PAHs and PAH4 contents. Clarify it.
*Discussion: The health risk assessment performed indicates that there is no risk concerning the consumption of seasoning flour products. However, in this part you can discuss that although the consumption of these kinds of products do not imply a health risk, the co-exposure to PAHs and long-term exposure could produce a health risk higher than the commented here.
*Line 353: In general, specify P-values obtained to affirm that there were or not significant differences, in this section or in result section
*Line 583: I think there is a mistake in the references 59 and 60
Author Response
Response to Reviewer 2
Dear Reviewer,
We sincerely appreciate your insightful and valuable comments and suggestions. All of your comments and suggestions are of great help to the improvement of the quality of our manuscript. Here we submit a new version of our revised manuscript according to the suggestions, titled “Contamination and health risk assessment of polycyclic aromatic hydrocarbons in seasoning flour products in Hunan, China” (ID: ijerph-2126415). You will see the difference made to the revised manuscript.
If you have any questions about this paper, please do not hesitate to contact me.
Sincerely yours,
Hong Qin
Extensive revisions to the manuscript have been marked up using the “Track Changes” function. Please find below an itemized list of POINT-BY-POINT responses to these comments:
Reviewer #2
Comments and Suggestions for Authors
The article entitled “Contamination and health risk assessment of polycyclic aromatic hydrocarbons in seasoning flour products in Hunan, China” describes an analytical methodology to analyse 16 priority PAHs in seasoning flour products. In addition, an exhaustive risk evaluation was performed considering the concentrations obtained and the consumption pattern for three ranges of age, children, adolescents and adults.
As authors confirm, there are only 2 articles dedicated to analysing PAHs in this kind of products, but these articles evaluated only a small number of PAHs. The number of analysed samples is big enough to be considered a representative study and the high detection rate (96.20%) highlights the importance of controlling these food samples. In my opinion, the present article has novelty enough to be published in International Journal of Environmental Research and Public Health. However, there are some minor changes that must be addressed before publication:
General commentary: As the author mentioned, there is scarce information about PAHs in these food products. Thus, the novelty of this work should be highlighted along the manuscript.
Rev: Thanks a lot for the valuable suggestion. We highlighted the novelty of our present study in the revised manuscript accordingly.
Line 77-78 in the revised manuscript: “Therefore, this study intends to conduct a preliminary investigation into this significant but still unclear field.”
Line 81-85 in the revised manuscript: The present study combined the detection results of 16 PAHs and consumption survey results to quantify the daily dietary exposure level of PAHs, and further assess the Margin of exposure (MOE) and the incremental lifetime cancer risk (ILCR) of the Hunan population for long-term consuming seasoning flour products to exposure PAHs, which is novel but of great significance.”
Line 552-553 in the revised manuscript: “This is beneficial to fill the gap in the research on the health risks of dietary exposure of PAHs in seasoning flour products.”
Line 50: I recommend putting a range of age instead put “millennials and post-millennials”
Rev: Thanks for your suggestion. We have changed “millennials and post-millennials” to the “population between the ages of 10 and 40” (Line 60 in the revised manuscript).
Line 53: The reference 14 (Investigation and analysis of market latiao food related indexes) could not be found. I recommend to authors to find another reference which support their affirmation
Rev: Thank you for pointing it out. The reference 14 (Investigation and analysis of market latiao food related indexes; doi:10.19754/j.nyyjs.20190915012) is a Chinese literature. We can download it from China National Knowledge Internet. However, considering that it could not be found from the literature retrieval database such as PubMed, Web of Science and Google Scholar et al., we have removed it from the revised manuscript.
Line 70-71: The same objective was described in lines 75-79. Merge both sentences
Rev: Thank you for your suggestion. We have adjusted these sentences.
Line 85-93 in the revised manuscript: “On the one hand, the cancer risk assessment of the Hunan population is conducive to exploring the potential health risk of PAHs dietary exposure caused by consuming seasoning flour products to the whole population, which contributes to providing a scientific basis for the formulation of maximum allowable daily intake and maximum levels of PAHs in seasoning flour products. On the other hand, quantitative description and difference analysis of PAHs pollution in various seasoning flour products were carried out to provide a reference for optimizing the processing process of food products factories and the purchase choice of consumers.”
Line 90: I understand you crush the samples prior to the extraction process. If yes, please specify. On the other hand, what’s the matter to make a salting-out QuEChERS? I believe that samples called “spicy strip” are solid samples with a minimal water content, so you will have only the organic phase, you do not need to separate any phase. Correct me if I am wrong, please.
Rev: Q1: Thanks for your advice. We have added the sample crushing operation.
Line 126 in the revised manuscript: “The seasoning flour product samples were crushed and mixed with grinder.”
Q2: Thank you for your question. In fact, with the exception of crispy seasoning flour products, most seasoning flour products have a water content of 20-30%. During extraction, NaCl and anhydrous MgSO4 can absorb the water of samples, which was conducive to the transfer of the tested substance to the organic phase.
Line 91: Change “NaCL” for “NaCl”. Besides, it was not previously abbreviated.
Rev: Thank you for your suggestion. We have revised it to “sodium chloride (NaCl)” accordingly (Line 128 in the revised manuscript).
Line 95: When you use cold centrifugation specify the temperature used.
Rev: Thank you for your suggestion. We have added the cold centrifugal temperature (4℃) in the revised manuscript (Line 132 in the revised manuscript).
Line 98: Eliminate “And”
Rev: We have deleted it accordingly (Line 134 in the revised manuscript).
Line 100: Replace “2min” for “2 min”
Rev: Thank you for pointing it out. We have corrected this mistake (Line 137 in the revised manuscript).
Line 128: Include the questionnaire questions in supplementary material
Rev: Thank you for your suggestion. The questionnaire questions have been added to supplementary material.
Line 259: You said that for all seasoning products, contain was from high to low in the order of filamentous samples, rod-shaped samples, flaky samples and granular samples. However, in the next lines you said that rod-shaped samples and flaky samples concentrations were higher than filamentous samples for total PAHs and PAH4 contents. Clarify it.
Rev: Thank you for pointing it out. We have corrected this mistake.
Line 384-386 in the revised manuscript: “In all kinds of seasoning flour products with different shapes, the contents of total PAHs, BaP and PAH4 from high to low were in the order of flaky samples, rod-shaped samples, filamentous samples, and granular samples.”
Discussion: The health risk assessment performed indicates that there is no risk concerning the consumption of seasoning flour products. However, in this part you can discuss that although the consumption of these kinds of products do not imply a health risk, the co-exposure to PAHs and long-term exposure could produce a health risk higher than the commented here.
Rev: Thank you for your suggestion. We have added a discussion of co-exposure and long-term to PAHs to the Discussion Section.
Line 535-537 in the revised manuscript: “Despite the health risks were tolerable, the co-exposure and long-term exposure to PAHs might produce higher health risks than those commented here. Therefore, the susceptible populations, such as children, and adolescents are recommended to limit the intake of seasoning flour products.”
Line 353: In general, specify P-values obtained to affirm that there were or not significant differences, in this section or in result section.
Rev: Thank you for your suggestion. We have specified P<0.05 as a marker of statistical difference (Line 497 in the revised manuscript).
Line 583: I think there is a mistake in the references 59 and 60.
Rev: Thank you for pointing it out. We have corrected this mistake.

Round 2
Reviewer 1 Report
Dear, regarding the revised manuscript submitted for review, I think the authors significantly improved the manuscript in forms that is acceptable for publication.